# OUT-OF-DISTRIBUTION DETECTION WITH HYPERSPHERICAL ENERGY

## ABSTRACT

The ability to detect if inputs are out-of-distribution (OOD) is essential to guarantee the reliability and safety of machine learning models that are deployed in an open environment. Recent studies have shown that an energy-based score is effective. However, unconstrained energy scores from a model trained with cross-entropy loss may not necessarily reflect the log-likelihood. To address this limitation, we introduce a novel *hyperspherical energy score* that connects energy with hyperspherical representations. By modeling hyperspherical representations using von Mises-Fisher distribution, our method provides a theoretical interpretation from a log-likelihood perspective, making it an effective and rigorous OOD detection indicator. Our method consistently achieves competitive performance on popular OOD detection benchmarks. On the large-scale ImageNet-1k benchmark, our method is more than 10 times faster than the KNN-based score, while simultaneously reducing the average FPR95 by 11.85%.

## 1 INTRODUCTION

The problem of out-of-distribution (OOD) detection has gained significant attention in the field of machine learning, particularly in the context of deep learning (Yang et al., 2021b). OOD detection refers to the task of identifying test samples that do not belong to the same distribution as the training data, which can occur due to various factors such as the natural shift in the model's operating environment. The ability to detect OOD samples is crucial for ensuring the reliability and safety of machine learning models (Amodei et al., 2016), especially in real-world applications where the consequences of misidentifying OOD samples can be severe.

A fundamental dilemma in OOD detection is that a discriminative classifier is typically trained to estimate the posterior probability $p(y|\mathbf{z})$ for class $y$ given an embedding $\mathbf{z}$, but lacks the explicit likelihood estimation of $p(\mathbf{z})$ needed for OOD detection. While many OOD scoring functions have been proposed for classification models, most of them cannot be rigorously interpreted as log-likelihood. Recent studies have shown that energy-based score (Liu et al., 2020) is a promising approach for OOD detection, which is defined as the negative log partition function of the softmax. However, Liu et al. (2020) employ the model trained with cross-entropy (CE) loss, which *produces unconstrained energy scores that do not always reflect the log-likelihood* (see Section 2 for theoretical justification).

To reconcile the dilemma, this paper proposes **Hyperspherical Energy** for OOD detection, a novel framework that first establishes the connection between the hyperspherical representations and Helmholtz free energy. Our framework mitigates the drawbacks of the previous energy-based approach and can be rigorously reasoned from a log-likelihood perspective for OOD detection. Specifically, hyperspherical energy operates on the latent embeddings of a discriminative classifier, which are modeled as a mixture of hyperspherical embeddings with constant norm (see Figure 1). Under such a probabilistic model, each class $c$ has a mean vector $\boldsymbol{\mu}_c$ in the hypersphere, where the class-conditional density function decays from the center exponentially by the similarity between an embedding $\mathbf{z}$ to the center: $\boldsymbol{\mu}_c^\top \mathbf{z}$. The hyperspherical energy function is defined in terms of the negative log partition function of the Gibbs-Boltzmann distribution:

$$E(\mathbf{z}) = -\tau \cdot \log \sum_{c=1}^{C} \exp(-E(\mathbf{z}, c)/\tau), \tag{1}$$

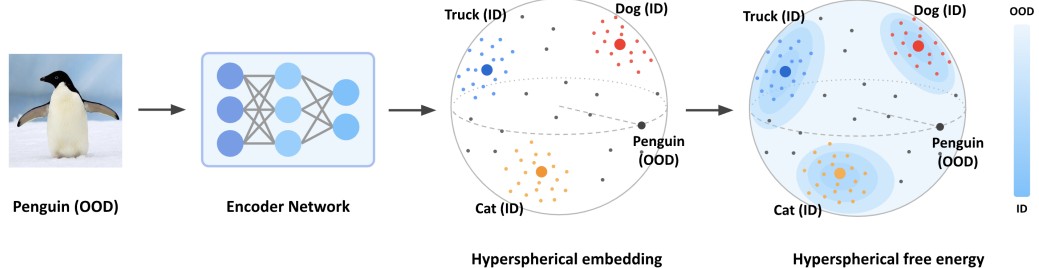

Figure 1: Overview of *hyperspherical energy* for OOD detection. The neural network is trained on the in-distribution (ID) data, which lies on the unit hypersphere in the latent space. The hyperspherical energy is theoretically equivalent to the negative log-likelihood $-\log p(\mathbf{z})$, which suits OOD detection.

where interestingly, $E(\mathbf{z}, c) = -\boldsymbol{\mu}_c^\top \mathbf{z}$ under our probabilistic model. *Theoretically, we show that hyperspherical energy $E(\mathbf{z})$ is equivalent to the negative log-likelihood score $-\log p(\mathbf{z})$ (up to some constant difference), and thus can act nicely as a density-based score for OOD detection.* Though simple and elegant in hindsight, establishing this connection was non-trivial. We lay out the full theoretical derivation in Section 3.1, along with how to effectively optimize for the hyperspherical energy in the context of modern neural networks (Section 3.2).

**Significance to the field.** We believe the research community can benefit from this new perspective of using hyperspherical energy for OOD detection. To highlight the advantages, we would like to draw the following contrast:

- *Hyperspherical energy vs. energy:* The key distinction between the two methods lies in the training-time objective and the resulting geometry of the embedding space, from which the energy score is derived. Liu et al. (2020) employ a cross-entropy loss and derive energy from the unconstrained Euclidean space. In contrast, our framework explicitly models the latent representations of the discriminative classifier as constrained hyperspherical distributions, under which the hyperspherical energy score is theoretically equivalent to negative log-likelihood score (up to some constant difference). Empirically on CIFAR-100, our method reduces the average FPR95 by ↓**40.45%**, which validates that the proposed hyperspherical energy score is significantly more effective than CE-based energy (Liu et al., 2020). Overall, our approach demonstrates notable strengths in both empirical performance and theoretical underpinnings.

- *Hyperspherical energy vs. non-parametric density estimation:* From the test-time perspective, one could also view our method as a parametric OOD scoring function, which stands in contrast with non-parametric density estimation. The current high-performing methods, such as KNN+ (Sun et al., 2022b) and CIDER (Ming et al., 2023), rely on a non-parametric KNN score, which requires a nearest neighbor search in the embedding space. Although recent studies on approximated nearest neighbor search have shown promise (Aumüller et al., 2020), the computation required increases with the size of the embedding pool $N$. In contrast, the hyperspherical energy score only incurs complexity $O(C)$, where the number of classes $C \ll N$ is independent of ID sample size. On a large-scale ImageNet benchmark, our method is more than **10x** faster than the KNN-based score, while simultaneously reducing the average FPR95 by ↓**11.85%** and establishing state-of-the-art performance.

## 2 PRELIMINARIES

**Out-of-distribution detection.** We begin by defining in-distribution (ID) data in the context of multi-class classification. We denote $\mathcal{X}$ as the input space, and $\mathcal{Y} = \{1, 2, \ldots, C\}$ represents the set of ID labels. The training set $\mathcal{D}_{\text{in}} = \{(\mathbf{x}_i, y_i)\}_{i=1}^N$ is sampled *i.i.d.* from the joint distribution $P_{\mathcal{X}\mathcal{Y}}$. The marginal distribution on $\mathcal{X}$ is denoted as $P_{\text{in}}$, and represents the distribution of ID data.

In the context of multi-class classification problems, out-of-distribution (OOD) data refers to data that belongs to an irrelevant distribution whose label set is disjoint from $\mathcal{Y}$. Such data should not be predicted by the model. The goal of OOD detection is to design an algorithm that can predict whether a test sample $\mathbf{x} \in \mathcal{X}$ belongs to the ID distribution ($y \in \mathcal{Y}$) or the OOD distribution ($y \notin \mathcal{Y}$).

**Energy-based OOD detection.** Recent studies have shown that energy score (Liu et al., 2020) is a promising approach for OOD detection, which builds upon a connection between an energy-based model (EBM) and discriminative classifiers. An EBM (Lecun et al., 2006) captures the compatibility between an input, $\mathbf{x}$ and state $y \in \mathcal{Y}$, using a non-probabilistic scalar energy function $E(\mathbf{x}, y)$. By treating the energies as unnormalized negative log probabilities, one can form a probability density by marginalizing over all possible $y' \in \mathcal{Y}$ through the Gibbs-Boltzmann distribution

$$p(y \mid \mathbf{x}) = \frac{\exp(-E(\mathbf{x}, y)/\tau)}{\int_{y' \in \mathcal{Y}} \exp(-E(\mathbf{x}, y')/\tau)}, \tag{2}$$

where $\tau$ denotes a temperature parameter. The Helmholtz free energy is defined as the negative log of the partition function.

$$E(\mathbf{x}) = -\tau \cdot \log \int_{y' \in \mathcal{Y}} \exp(-E(\mathbf{x}, y')/\tau). \tag{3}$$

Liu et al. (2020) introduced an energy score for OOD detection that builds upon a connection between EBMs and discriminative models. Specifically, the predictive probability distribution can be derived using a softmax function over the logits of a discriminative neural classifier:

$$p(y = c \mid \mathbf{x}) = \frac{\exp(f_c(\mathbf{x})/\tau)}{\sum_{j=1}^{C} \exp(f_j(\mathbf{x})/\tau)} = \frac{p(\mathbf{x}, y = c)}{p(\mathbf{x})}, \tag{4}$$

where $f_c(\mathbf{x})$ denotes the logit corresponding to the class $c$. This allows the authors to define the free energy score as the denominator of the softmax function:

$$E(\mathbf{x}; f) = -\tau \cdot \log \sum_{j=1}^{C} \exp(f_j(\mathbf{x})/\tau), \tag{5}$$

which is used to differentiate ID vs. OOD. An input is considered to be ID if the energy score is low, and vice versa.

**Limitation of Liu et al. (2020).** The energy score derived from the Euclidean space logits $f(\mathbf{x})$ can be *unconstrained*, and does not necessarily reflect the log-likelihood. To see this, one can rewrite the posterior probability $p(y = c|\mathbf{x}) = p(\mathbf{x}, y = c)/p(\mathbf{x})$. Note that one can always rescale the logits $f(\mathbf{x})$ by a function $\log g(\mathbf{x})$, without changing the posterior probability: $p(y = c|\mathbf{x}) = p(\mathbf{x}, y = c)g(\mathbf{x})/p(\mathbf{x})g(\mathbf{x})$. In this case, the negative log of the denominator in softmax (i.e., the energy score) is $-\log[p(\mathbf{x})g(\mathbf{x})]$, which is no longer truthfully reflecting the $-\log p(\mathbf{x})$.

## 3 HYPERSPHERICAL ENERGY FOR OOD DETECTION

We propose *Hyperspherical Energy* for OOD detection, a novel framework that first connects energy with learning hyperspherical representations. Our framework mitigates the drawbacks of the previous energy-based approach and can be rigorously interpreted from a log-likelihood perspective. In what follows, we first provide a definition of hyperspherical energy (Section 3.1) and then show how optimized hyperspherical space can be used for distinguishing ID and OOD data (Section 3.2).

### 3.1 HYPERSPHERICAL ENERGY

In this section, we introduce the proposed notion of hyperspherical energy and theoretically justify its feasibility as an indicator function of OOD. The key insight is that hyperspherical energy is directly linked to the log-likelihood of the embedding, which can be used as a measure of how well a given embedding matches the distribution of ID samples. We start by introducing the necessary background on hyperspherical embeddings.

**Modeling hyperspherical space.** Geometrically, hyperspherical embeddings are located on the surface of a hypersphere. A hypersphere is a topological space that is equivalent to a standard $(d-1)$-sphere, which represents the set of points in $d$-dimensional Euclidean space that are *equidistant* from the center (see Figure 1). The unit hypersphere is a special sphere with a unit radius and is denoted as $S^{d-1} := \{\mathbf{z} \in \mathbb{R}^d | \|\mathbf{z}\|_2 = 1\}$ in the $(d-1)$-dimensional space.

The hyperspherical embeddings can be modeled by the von Mises-Fisher (vMF) distribution, a well-known distribution in directional statistics (Jupp & Mardia, 2009). For a unit vector $\mathbf{z} \in \mathbb{R}^d$ in class $c$, the probability density function of the vMF distribution is defined as

$$p(\mathbf{z} \mid y = c) = Z_d(\kappa) \exp(\kappa \boldsymbol{\mu}_c^\top \mathbf{z}), \tag{6}$$

where $\boldsymbol{\mu}_c \in \mathbb{R}^d$ denotes the mean direction of the class $c$, $\kappa \geq 0$ denotes the concentration of the distribution around $\boldsymbol{\mu}_c$, and $Z_d(\kappa)$ denotes the normalization factor. Under this probabilistic model, an embedding $\mathbf{z}$ is assigned to the class $c$ with the following normalized probability

$$
\begin{aligned}
p(y = c \mid \mathbf{z}; \{\kappa, \boldsymbol{\mu}_j\}_{j=1}^C) &= \frac{Z_d(\kappa) \exp(\kappa \boldsymbol{\mu}_c^\top \mathbf{z})}{\sum_{j=1}^C Z_d(\kappa) \exp(\kappa \boldsymbol{\mu}_j^\top \mathbf{z})} \\
&= \frac{\exp(\boldsymbol{\mu}_c^\top \mathbf{z}/\tau)}{\sum_{j=1}^C \exp(\boldsymbol{\mu}_j^\top \mathbf{z}/\tau)},
\end{aligned} \tag{7}
$$

where $\tau = 1/\kappa$ denotes a temperature parameter.

**Defining and interpreting hyperspherical energy.** Now, we are ready to draw a novel connection between the hyperspherical distribution and the Helmholtz free energy, which has not been explored in the past. In particular, Eq. 7 can be interpreted as the Gibbs-Boltzmann distribution in Eq. 2, where $E(\mathbf{z}, y = c) = -\boldsymbol{\mu}_c \cdot \mathbf{z}$ measures the compatibility between an input embedding $\mathbf{z}$ and a class prototype $\boldsymbol{\mu}_c$. A compatible input-output pair should result in a lower energy value and vice versa. Accordingly, we can define the hyperspherical energy function over $\mathbf{z} \in \mathbb{R}^d$ in terms of the negative log partition function (or denominator in Eq. 7):

$$E(\mathbf{z}; \{\tau, \boldsymbol{\mu}_j\}_{j=1}^C) = -\tau \cdot \log \sum_{j=1}^C \exp(\boldsymbol{\mu}_j^\top \mathbf{z}/\tau). \tag{8}$$

Our next lemma shows that hyperspherical energy can be functionally equivalent to the log-likelihood score, for OOD detection purposes.

**Lemma 3.1.** *The hyperspherical free energy $E(\mathbf{z})$ is a negative logarithmic function of the likelihood $p(\mathbf{z})$, with a constant difference. The two measurements return the same level set for OOD detection.*

*Proof.* The likelihood $p(\mathbf{z})$ is the probability of observing a given embedding $\mathbf{z}$ under the vMF distribution. $p(\mathbf{z})$ can be calculated as a summation of class-conditional likelihood $p(\mathbf{z} \mid y = c)$, weighted by the class prior $p(y = c)$:

$$p(\mathbf{z}) = \sum_{j=1}^C p(\mathbf{z} \mid y = c) p(y = c) = \frac{Z_d(\kappa)}{C} \sum_{j=1}^C \exp(\boldsymbol{\mu}_i^\top \mathbf{z}/\tau), \tag{9}$$

where the right-hand side is obtained by plugging in Eq. 6 and class prior probability $1/C$.

By connecting the hyperspherical energy $E(\mathbf{z})$ in Eq. 8 and the likelihood $p(\mathbf{z})$ in Eq. 9, we can show the following equivalence between the two:

$$E(\mathbf{z}; \{\tau, \boldsymbol{\mu}_j\}_{j=1}^C) = -\tau \cdot \underbrace{\log p(\mathbf{z})}_{\text{log likelihood of } \mathbf{z}} - \underbrace{\tau \cdot \log \frac{C}{Z_d(1/\tau)}}_{\text{const, independent of } \mathbf{z}} \tag{10}$$

Note that the second term is independent of $\mathbf{z}$, and can be treated as a constant. Thus, hyperspherical energy $E(\mathbf{z})$ is functionally equivalent to the negative log-likelihood score, for OOD detection.

**Hyperspherical energy for OOD detection.** Given the above equivalence between hyperspherical energy and negative log-likelihood, we propose the following OOD indicator function:

$$S(\mathbf{z}) = -E(\mathbf{z}; \{\tau, \boldsymbol{\mu}_j\}_{j=1}^C). \tag{11}$$

Here we negate the sign of hyperspherical energy to align with the convention that positive samples (ID) have higher scores. To make a decision, we use the level set: $G_\lambda(\mathbf{x}) = \mathbb{1}\{S(\mathbf{z}) \geq \lambda\}$, where samples with higher scores $S(\mathbf{z})$ are classified as ID, and vice versa. The threshold $\lambda$ is chosen based on the ID score at a certain percentile (e.g., 95%).

### 3.2 Optimizing Hyperspherical Energy in Deep Neural Networks

So far, we have established that hyperspherical energy derived under the vMF distribution is desirable for OOD detection. Now, we discuss how to optimize neural networks for hyperspherical energy.

**Loss function.** The goal is to train the neural network, such that the hyperspherical embedding conforms to a mixture of vMF distributions defined in Eq. 7. Specifically, we consider a deep neural network $h : \mathcal{X} \mapsto \mathbb{R}^d$ which encodes an input $\mathbf{x} \in \mathcal{X}$ to a normalized feature embedding $\mathbf{z} := h(\mathbf{x})$, on the hypersphere. To optimize for the vMF distribution in the embedding space, one can directly perform the maximum likelihood estimation on the training dataset $\{(\mathbf{x}_i, y_i)\}_{i=1}^{N}$:

$$\mathrm{argmax}_\theta \prod_{i=1}^{N} p(y = y_i \mid \mathbf{z}_i; \{\kappa, \boldsymbol{\mu}_j\}_{j=1}^{C}), \tag{12}$$

where $i$ is the index of the sample, $j$ is the index of class, and $N$ is the size of the training set. In effect, this loss encourages each ID sample to have a high probability assigned to the correct class in the mixtures of the vMF distributions. While hyperspherical learning algorithms have been studied (Mettes et al., 2019; Du et al., 2022a; Ming et al., 2022), none of the works explored its connection to hyperspherical energy in test-time OOD detection, which is our distinct focus. We discuss the differences further in Section 3.3. Moreover, we show news insights below that minimizing this loss function effectively reduces the hyperspherical energy for ID data.

**How does the loss function shape hyperspherical energy?** By taking negative-log likelihood, the objective function in Eq. 12 is equivalent to minimizing the following loss:

$$\mathcal{L} = \mathbb{E}_{(\mathbf{x},y)\in\mathcal{D}_{\mathrm{in}}} - \log p(y = y \mid \mathbf{z}; \{\tau, \mu_j\}_{j=1}^{C}). \tag{13}$$

We show that our learning objective can induce lower hyperspherical energy for ID data. To see this, we can rewrite the loss function in terms of hyperspherical energy by expanding Eq. 13 with the probability $p(y = y \mid \mathbf{z})$ derived in Eq. 7:

$$\mathcal{L} = \mathbb{E}_{(\mathbf{x},y)\in\mathcal{D}_{\mathrm{in}}} \left( -\log \frac{\exp(\boldsymbol{\mu}_y^\top \mathbf{z}/\tau)}{\sum_{j=1}^{C} \exp(\boldsymbol{\mu}_j^\top \mathbf{z}/\tau)} \right)$$

$$= \mathbb{E}_{(\mathbf{x},y)\in\mathcal{D}_{\mathrm{in}}} \left( \frac{1}{\tau} E(\mathbf{z}, y) + \log \sum_{j=1}^{C} \exp(-E(\mathbf{z}, j)/\tau) \right) \tag{14}$$

During optimization, the loss reduces the hyperspherical energy for the correct class, while increasing the hyperspherical energy for other classes. This can be justified by analyzing the gradient of the loss function with respect to the model parameters $\theta$, for an embedding $\mathbf{z}$ and its associated class label $y$:

$$\frac{\partial \mathcal{L}(\mathbf{z}, y; \theta)}{\partial \theta} = \frac{1}{\tau} \frac{\partial E(\mathbf{z}, y)}{\partial \theta} - \frac{1}{\tau} \sum_{j=1}^{C} \frac{\partial E(\mathbf{z}, j)}{\partial \theta} \cdot \frac{\exp(-E(\mathbf{z}, j)/\tau)}{\sum_{c=1}^{C} \exp(-E(\mathbf{z}, c)/\tau)}$$

$$= \frac{1}{\tau} (\underbrace{\frac{\partial E(\mathbf{z}, y)}{\partial \theta}(1 - p(Y = y \mid \mathbf{z}))}_{\downarrow \text{ for } y} - \underbrace{\sum_{j \neq y} \frac{\partial E(\mathbf{z}, j)}{\partial \theta} p(Y = j \mid z))}_{\uparrow \text{ for } j \neq y} \tag{15}$$

The sample-wise hyperspherical energy $E(\mathbf{z})$ of ID data is $E(\mathbf{z}) = -\tau \cdot \log \sum_{c=1}^{C} \exp(-E(\mathbf{z}, c)/\tau)$, which is dominated by the $E(\mathbf{z}, y)$ with ground truth label. Hence, the training overall induces lower hyperspherical energy for ID data. By further connecting to our Lemma 3.1, this low hyperspherical energy can directly translate into high ($\uparrow$) log-likelihood $\log p(\mathbf{z})$, for training data distribution.

### 3.3 Differences with Existing Approaches

Table 1 summarizes the key distinctions among prevalent hyperspherical methodologies, specifically in terms of their training-time loss functions and test-time scoring functions. Both SSD+ (Sehwag et al., 2021) and KNN+ (Sun et al., 2022b) employ the SupCon (Khosla et al., 2020) loss during training, which does not explicitly model the latent representations as vMF distributions. Instead of

Table 1: Summary of key distinction among different hyperspherical approaches.

| | Training-time loss function | Test-time scoring function |
|---|---|---|
| SSD+ | SupCon | Mahalanobis (parametric) |
| KNN+ | SupCon | KNN (non-parametric) |
| CIDER | vMF | KNN (non-parametric) |
| SIREN | vMF | KNN or $\max p(y\|\mathbf{z})$ |
| Hyperspherical energy (ours) | vMF | Hyperspherical energy (parametric) |

promoting instance-to-prototype similarity, `SupCon` promotes instance-to-instance similarity among positive pairs. `SupCon`'s loss formulation thus does not directly correspond to the vMF distribution. Geometrically speaking, our framework directly operates on the vMF distribution, a key property that enables our hyperspherical energy with log-likelihood interpretation. `CIDER` (Ming et al., 2023) and our approach differ significantly in terms of OOD scoring function formulation in testing time. Our method establishes the novel connection between the hyperspherical representations and Helmholtz free energy for OOD detection, which enjoys rigorous theoretical interpretation from a log-likelihood perspective, while `CIDER` does not. We show empirically in Section 4 that our proposed OOD score achieves competitive performance on different OOD detection benchmarks, and is much more computationally efficient by eliminating the need for time-consuming KNN searches.

## 4 EXPERIMENTS

This section presents empirical experiments on several benchmark datasets to validate the effectiveness of our proposed hyperspherical energy OOD detection method. In particular, we aim to show that the hyperspherical energy score (1) outperforms the original energy score, (2) achieves state-of-the-art performance, and (3) requires less computation than the current best methods. Code and data will be released publicly for reproducible research.

### 4.1 SETUP

**Datasets.** We consider CIFAR-10, CIFAR-100 (Krizhevsky, 2009), and ImageNet-1k (Deng et al., 2009) as ID datasets, which are commonly used in literature. We then evaluate the methods on five common OOD datasets: SVHN (Netzer et al., 2011), `Places365` (Zhou et al., 2016), LSUN (Yu et al., 2016), `iSUN` (Xu et al., 2015), and `Textures` (Cimpoi et al., 2014). For large-scale ImageNet evaluation, we follow the same setup as in Sun et al. (2022b).

**Evaluation metrics.** We utilize the following metrics to evaluate the performance of our method: (1) false positive rate (FPR95) of OOD samples when the true positive rate of ID samples is set at 95%, (2) area under the receiver operating characteristic curve (AUROC), and (3) ID classification accuracy (ID ACC). These metrics are widely used to assess the efficacy of OOD detection methods.

**Experiment details.** We include full details of model training in Appendix A. For OOD detection evaluation, we use our proposed hyperspherical energy score with test-time temperature $\tau_{\text{test}} = 0.05$ based on validation (see Appendix B). We further show the effect of temperature in our ablation study (Section 4.3). For ID classification evaluation, we follow the standard linear probing protocol and train a linear classifier on top of the penultimate layer features. The results are provided in Appendix C.

### 4.2 MAIN RESULTS

**Hyperspherical energy score outperforms energy score.** We begin by highlighting the improvement of the hyperspherical energy score over the original energy score defined in Eq. 5. Figure 2 illustrates a comparison of the performance of `Energy` and `Hyperspherical Energy` on CIFAR-100. Our method reduces the average FPR95 by ↓**40.45**%, which validates that the proposed hyperspherical energy score is significantly more effective for OOD detection. Our performance improvement can also be reasoned theoretically, as we show in Lemma 3.1. Different from Liu et al. (2020), hyperspherical energy operates on the latent representations with vMF distributions, which enables a rigorous interpretation from a log-likelihood perspective. Overall, our method enjoys both strong empirical performance and theoretical justification. Additionally, for a more detailed understanding, we have included ID-OOD density visualizations in Appendix F.

Table 2: Average and standard deviation test-time compute time per sample across samples in each ID dataset. The hyperspherical energy score is computationally more efficient compared to the KNN search.

| Score | Score compute time ($\mu$s$\downarrow$) | | | |
|---|---|---|---|---|
| | CIFAR-10 | CIFAR-100 | ImageNet ($\alpha = 1\%$) | ImageNet ($\alpha = 100\%$) |
| KNN+ or CIDER | $3.5619 \pm 0.2646$ | $5.4886 \pm 0.1854$ | $3.6157 \pm 0.1186$ | $336.1780 \pm 4.3871$ |
| Hyperspherical energy | $\mathbf{0.0076} \pm 0.0003$ | $\mathbf{0.0157} \pm 0.0007$ | $\mathbf{0.2745} \pm 0.0145$ | $\mathbf{0.2745} \pm 0.0145$ |

Table 3: OOD detection performance for CIFAR-100 (ID) with ResNet-34. Hyperspherical energy achieves competitive performance with state-of-the-art methods.

| Method | OOD Dataset | | | | | | | | | | Average | |
|---|---|---|---|---|---|---|---|---|---|---|---|---|
| | SVHN | | Places365 | | LSUN | | iSUN | | Texture | | | |
| | FPR$\downarrow$ | AUROC$\uparrow$ | FPR$\downarrow$ | AUROC$\uparrow$ | FPR$\downarrow$ | AUROC$\uparrow$ | FPR$\downarrow$ | AUROC$\uparrow$ | FPR$\downarrow$ | AUROC$\uparrow$ | FPR$\downarrow$ | AUROC$\uparrow$ |
| Methods using cross-entropy loss | | | | | | | | | | | | |
| MSP | 83.64 | 77.21 | 84.14 | 75.18 | 80.12 | 78.88 | 84.20 | 74.99 | 87.38 | 72.56 | 83.90 | 75.76 |
| ODIN | 78.27 | 83.59 | 83.33 | 76.05 | 70.75 | 85.00 | 77.95 | 80.38 | 88.37 | 73.16 | 79.73 | 79.64 |
| Mahalanobis | 78.92 | 79.16 | 91.59 | 66.64 | 97.70 | 58.42 | 75.53 | 78.44 | 59.24 | 82.86 | 80.60 | 73.10 |
| Energy | 77.02 | 83.97 | 83.47 | 75.92 | 68.79 | 85.37 | 76.76 | 80.71 | 88.97 | 72.96 | 79.00 | 79.79 |
| ViM | 49.04 | 90.80 | 84.58 | 73.70 | 94.10 | 71.51 | 43.45 | 91.01 | 49.49 | 88.85 | 64.13 | 83.17 |
| ReAct | 67.07 | 86.83 | 80.98 | 77.39 | 62.89 | 86.90 | 61.62 | 86.61 | 75.69 | 82.85 | 69.65 | 84.12 |
| DICE | 53.23 | 89.70 | 83.03 | 75.49 | 44.63 | 91.38 | 74.87 | 79.05 | 84.68 | 73.29 | 68.09 | 81.78 |
| SHE | 71.40 | 82.45 | 89.94 | 68.54 | 67.04 | 84.47 | 85.82 | 73.37 | 89.08 | 67.87 | 80.66 | 75.34 |
| FeatureNorm | 52.69 | 87.95 | 95.26 | 55.62 | 5.96 | 98.74 | 99.33 | 38.51 | 62.11 | 76.16 | 63.07 | 71.40 |
| Methods using hyperspherical representations | | | | | | | | | | | | |
| SSD+ | 27.82 | 95.06 | 79.03 | 78.12 | 63.58 | 88.11 | 79.88 | 83.42 | 76.08 | 82.75 | 65.28 | 85.49 |
| KNN+ | 51.53 | 90.93 | 77.59 | **79.37** | 48.67 | 91.41 | 61.42 | 87.53 | 62.52 | 86.48 | 60.35 | 87.14 |
| CIDER | 23.06 | 95.16 | 80.08 | 73.10 | 16.18 | 96.34 | 72.04 | 80.54 | 45.39 | 90.07 | 47.35 | 87.04 |
| Hyperspherical energy | **17.81** | **96.39** | **76.74** | 76.01 | **8.46** | **98.25** | 57.92 | 85.98 | **31.83** | **93.24** | **38.55** | **89.97** |

**Hyperspherical energy is computationally much faster than KNN-based methods.** The current state-of-the-art methods, such as `KNN+` and `CIDER`, rely on a non-parametric KNN score, which requires a nearest neighbor search in the embedding space. Although recent studies on approximated nearest neighbor search have shown promise (Aumüller et al., 2020), the computation required increases with the size of the embedding pool $N$. In contrast, the hyperspherical energy score only incurs complexity $O(C)$, where the number of classes $C \ll N$ is independent of ID sample size. Thus, our method can significantly reduce computation overhead. To demonstrate this, we conduct an experiment to compare the average compute time between the KNN score and the hyperspherical energy score. In this experiment, we assess the performance of the model

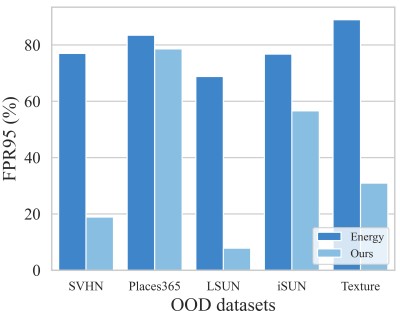

Figure 2: Performance of Energy vs. Hyperspherical Energy on CIFAR-100 (ID) with ResNet-34. Our method consistently outperforms Energy on different OOD datasets.

trained on CIFAR-10, CIFAR-100, and ImageNet (Deng et al., 2009) using default values of $k$ in (Sun et al., 2022b). In particular, we measure the time taken to calculate the score per sample, excluding the time to extract the sample embedding from the neural network. KNN-based methods utilize the Faiss library (Johnson et al., 2019), which is optimized for efficient nearest-neighbor search. The size of the embedding pool is 50,000 for both CIFAR-10 and CIFAR-100, 12K for ImageNet ($\alpha = 1\%$), and 1.2M for ImageNet ($\alpha = 100\%$), where $\alpha$ denotes the fraction of training data used for the nearest neighbor search (Sun et al., 2022b; Ming et al., 2023). Our results, shown in Table 2, demonstrate that hyperspherical energy is significantly faster than the KNN-based methods, making it a computationally desirable alternative for OOD detection tasks.

**Hyperspherical energy outperforms current state-of-the-art method.** In this section, we compare the performance of `Hyperspherical Energy` to that of several state-of-the-art methods. The compared methods are categorized as either hyperspherical-based or non-hyperspherical-based. All the non-hyperspherical-based methods, including `MSP` (Hendrycks & Gimpel, 2017), `ODIN` (Liang et al., 2018), `Mahalanobis` (Lee et al., 2018b), `Energy` (Liu et al., 2020), `ViM` (Wang et al., 2022), `ReAct` (Sun et al., 2021), `DICE` (Sun & Li, 2022), `SHE` (Zhang et al., 2023), and `FeatureNorm` (Yu et al., 2023b) use the softmax cross-entropy (CE) loss for model training. In particular, `ReAct` (Sun et al., 2021) improves the energy score by introducing a rectified activation, which reduces model overconfidence in OOD data. `DICE` (Sun & Li, 2022) utilizes logit sparsification to enhance the vanilla energy score. `SHE` computes the dot product between the sample embedding

Table 4: OOD detection performance for ImageNet (ID) with ResNet-50. Hyperspherical energy achieves competitive performance compared to state-of-the-art methods.

| Method | OOD Dataset | | | | | | | | Average | |
| | iNaturalist | | SUN | | Places | | Textures | | | |
| | FPR ↓ | AUROC ↑ | FPR ↓ | AUROC ↑ | FPR ↓ | AUROC ↑ | FPR ↓ | AUROC ↑ | FPR ↓ | AUROC ↑ |
|---|---|---|---|---|---|---|---|---|---|---|
| *Methods using cross-entropy loss* | | | | | | | | | | |
| MSP | 54.99 | 87.74 | 70.83 | 80.86 | 73.99 | 79.76 | 68.00 | 79.61 | 66.95 | 81.99 |
| ODIN | 47.66 | 89.66 | 60.15 | 84.59 | 67.89 | 81.78 | 50.23 | 85.62 | 56.48 | 85.41 |
| Mahalanobis | 97.00 | 52.65 | 98.50 | 42.41 | 98.40 | 41.79 | 55.80 | 85.01 | 87.43 | 55.47 |
| Energy | 55.72 | 89.95 | 59.26 | 85.89 | 64.92 | 82.86 | 53.72 | 85.99 | 58.41 | 86.17 |
| ViM | 71.85 | 87.44 | 82.06 | 81.05 | 83.31 | 78.38 | 14.95 | 96.81 | 63.04 | 85.92 |
| ReAct | 20.38 | 96.22 | **24.20** | **94.20** | **33.85** | **91.58** | 47.30 | 89.80 | 31.43 | 92.95 |
| DICE | 25.63 | 94.49 | 35.15 | 90.83 | 46.49 | 87.48 | 31.72 | 90.30 | 34.75 | 90.77 |
| SHE | 35.69 | 92.49 | 36.85 | 90.81 | 47.47 | 87.19 | 29.10 | 92.44 | 37.28 | 90.73 |
| FeatureNorm | 22.01 | 95.76 | 42.93 | 90.21 | 56.80 | 84.99 | 20.07 | 95.39 | 35.45 | 91.59 |
| *Methods using hyperspherical representation* | | | | | | | | | | |
| SSD+ | 57.16 | 87.77 | 78.23 | 73.10 | 81.19 | 70.97 | 36.37 | 88.52 | 63.24 | 80.09 |
| KNN+ ($\alpha = 100\%$) | 30.18 | 94.89 | 48.99 | 88.63 | 59.15 | 84.71 | 15.55 | 95.40 | 38.47 | 90.91 |
| KNN+ ($\alpha = 1\%$) | 30.83 | 94.72 | 48.91 | 88.40 | 60.02 | 84.62 | 16.97 | 94.45 | 39.18 | 90.55 |
| CIDER | 29.00 | 94.55 | 47.80 | 89.14 | 58.76 | 84.58 | 14.89 | 95.96 | 37.61 | 91.06 |
| Hyperspherical energy | **8.76** | **98.00** | 36.95 | 91.52 | 49.33 | 87.67 | **11.45** | **96.56** | **26.62** | **93.44** |

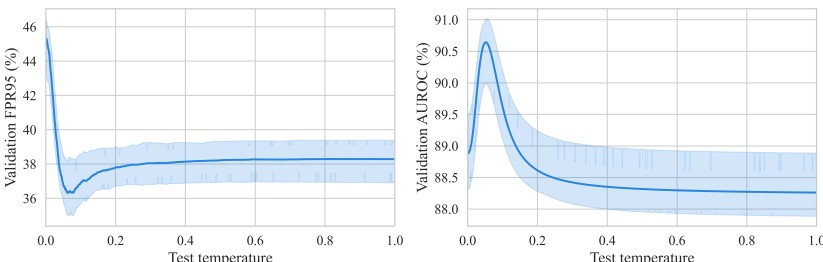

Figure 3: Ablation on test-time temperatures. The results are averaged across the 5 OOD test sets, based on ResNet-34 trained on CIFAR-100. Shades indicate variance across 3 independent training runs.

and the class mean of the predicted class. Hyperspherical-based methods, on the other hand, include `SSD+` (Sehwag et al., 2021), `KNN+` (Sun et al., 2022b), and `CIDER` (Ming et al., 2023). `SSD+` applies the Mahalanobis score (Lee et al., 2018b) on a model pre-trained with `SupCon` loss (Khosla et al., 2020). `KNN+` and `CIDER` both use nearest neighbor distance for the OOD score, but the model is trained with different contrastive losses. Further details about the training and evaluation of each baseline method are included in Appendix A.

As shown in Table 3, our method displays competitive OOD detection performance compared to existing ones. Here, we highlight two observations: (1) `Hyperspherical Energy` outperforms the current state-of-the-art method `CIDER`, by ↓**8.80**% in FPR95. While both methods are trained to optimize for vMF embeddings, our method differs fundamentally by the test-time OOD detection score. `CIDER` uses the KNN-based distance, which imposes the same computation burden as Sun et al. (2022b) (see Table 2). In contrast, hyperspherical energy reduces the computational complexity while providing a direct theoretical interpretation from a log-likelihood perspective. (2) `Hyperspherical Energy` generally outperforms energy-based scores derived from cross-entropy loss, including the improved variants `ReAct` and `DICE`. We additionally report the results for CIFAR-10 in Table 6 (Appendix).

### 4.3 ADDITIONAL ABLATIONS

**Hyperspherical energy is competitive on large-scale datasets.** To further examine the performance of our method on real-world tasks, we evaluate `Hyperspherical Energy` on the more challenging large-scale benchmarks. Specifically, we use ImageNet (Deng et al., 2009), consisting of 1,000 classes as an ID dataset. We fine-tune a pre-trained ResNet-50 model in Khosla et al. (2020) for 100 epochs, with an initial learning rate of 0.01 and cosine annealing schedule. For computational efficiency, we only update the weights of the last residual block and the nonlinear projection head, while freezing the parameters in the first three residual blocks. Following the ImageNet-based OOD detection benchmark (Huang & Li, 2021), we evaluate the model on four test OOD datasets that are subsets of `iNaturalist` (Van Horn et al., 2018), SUN (Xiao et al., 2010),

`Places` (Zhou et al., 2016) and `Textures` (Cimpoi et al., 2014). We set the test-time temperature for the hyperspherical energy score to 0.05. The results are presented in Table 4, which shows that `Hyperspherical Energy` performs competitively, achieving state-of-the-art results with an average FPR95 of 26.62%. We include Figure 4 (Appendix) to illustrate the learned embeddings via UMAP visualization (McInnes et al., 2018) on ImageNet.

**Ablation study on temperature $\tau$.** We conduct an ablation study on the test-time temperature parameter to investigate its impact on OOD detection performance, as shown in Figure 3. The curve highlights that the optimal value of $\tau_{\text{test}}$ roughly matches the training-time temperature. This also matches our theory, where hyperspherical energy under the same temperature $\tau$ in testing allows recovering the negative log-likelihood $-\log p(\mathbf{z})$ (up to some constant) learned in training time.

## 5 RELATED WORKS

**Out-of-distribution detection.** Out-of-distribution detection is an essential component for reliable machine learning systems that operate in the open world. Recent studies in OOD detection focus on train-time regularization and test-time scoring function (Yang et al., 2021b). In particular, train-time regularization approaches (Bevandić et al., 2018; Malinin & Gales, 2018; Geifman & El-Yaniv, 2019; Hein et al., 2019; Meinke & Hein, 2020; Mohseni et al., 2020; Van Amersfoort et al., 2020; Yang et al., 2021a; Du et al., 2022b; Wei et al., 2022; Katz-Samuels et al., 2022; Colombo et al., 2022; Yu et al., 2023a) design an algorithm to encourage the model to produce desired properties. For example, the model is regularized to produce lower confidence (Hendrycks et al., 2019; Lee et al., 2018a), higher energy (Du et al., 2022c; Liu et al., 2020), or desirable embedding space (Du et al., 2022a; Ming et al., 2023; Tao et al., 2023). On the other hand, a line of works also focuses on deriving test-time OOD scoring functions, including confidence-based scores (Hendrycks & Gimpel, 2017; Lakshminarayanan et al., 2017; Liang et al., 2018), distance-based score (Lee et al., 2018b; Tack et al., 2020; Sehwag et al., 2021; Sastry & Oore, 2020; Ren et al., 2021; Sun et al., 2022b; Du et al., 2022a; Ming et al., 2022; Ren et al., 2023), energy-based score (Liu et al., 2020; Wang et al., 2021; Morteza & Li, 2022; Zhang et al., 2023; Djurisic et al., 2023), gradient-based score (Huang et al., 2021), and multimodal score (Ming et al., 2022). Different from prior works, our method offers direct justification based on the likelihood view. Specifically, we regularize a model to produce hyperspherical representations aligned with the von Mises-Fisher distribution and propose hyperspherical energy as a test-time scoring function, which has not been explored in the past.

**Hyperspherical representation.** Hyperspherical representation under vMF distribution has been extensively used in various machine learning applications, such as supervised classification (Kobayashi, 2021; Scott et al., 2021; Govindarajan et al., 2023), face verification (Hasnat et al., 2017; Conti et al., 2022), generative modeling (Davidson et al., 2018), segmentation (Hwang et al., 2019; Sun et al., 2022a), and clustering (Gopal & Yang, 2014). In addition, some researchers have utilized the vMF distribution for anomaly detection by employing generative models and using it as the prior for zero-shot learning (Chen et al., 2020) and document analysis (Zhuang et al., 2017). Recent studies devise vMF-based learning for OOD detection, which offers a direct and clear geometrical interpretation of hyperspherical embeddings. Specifically, Du et al. (2022a) explores modeling representations into vMF distribution for object-level OOD detection, while `CIDER` (Ming et al., 2023) shows that additional regularization for large angular distances among different class prototypes is crucial for achieving strong OOD detection performance. Our work adopts a vMF-based learning approach for OOD detection and builds upon these recent studies by regularizing a model to produce representations that align with the vMF distribution and proposing a new hyperspherical energy scoring function for test-time OOD detection.

## 6 CONCLUSION AND OUTLOOK

In this paper, we introduce hyperspherical energy for detecting out-of-distribution samples. Our approach is based on a reformulation of energy score in the hyperspherical space, which enjoys a sound interpretation from a log-likelihood perspective. Through extensive experiments on common OOD detection benchmarks, we demonstrate the superior performance of our approach, including on large-scale detection tasks. Moreover, our method outperforms KNN-based approaches in terms of computational efficiency. Finally, we conduct an ablation study to evaluate the method's sensitivity under different test temperatures. Overall, our proposed hyperspherical energy score provides a promising solution for both effective and efficient OOD detection.

**Ethics statements.** OOD detection can be used to improve the reliability of safety-critical applications, which can have a significant positive impact on society. Moreover, *hyperspherical energy is a privacy-friendly OOD detection score.* Our OOD detection algorithm offers privacy benefits over the KNN score by not requiring access to a pool of ID data. In situations where data privacy is a concern, the use of the KNN score may not be feasible as it requires access to a certain amount of labeled data, which can pose privacy risks. Our algorithm utilizes hyperspherical energy to estimate uncertainty and detect OOD samples, only requiring prototypes of each class. This makes it a valuable tool in various applications, including healthcare (Murdoch, 2021), finance (Liu et al., 2021), and other industries where sensitive information needs to be protected. However, it is important to note that the usefulness and effectiveness of OOD detection may vary across different applications and domains, and careful evaluation and validation are necessary before its deployment.

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

# A  EXPERIMENTAL DETAILS

**Software and Hardware.**   We conduct our experiments NVIDIA RTX A6000 GPUs (48GB VRAM). We use Ubuntu 22.04.2 LTS as the operating system and install the NVIDIA CUDA Toolkit version 11.6 and cuDNN 8.9. All experiments are implemented in Python 3.8.1 using the PyTorch 1.12.1 framework.

**Training cross-entropy models.**   For methods using cross-entropy loss, such as MSP (Hendrycks & Gimpel, 2017), ODIN (Liang et al., 2018), Mahalanobis (Lee et al., 2018b), Energy (Liu et al., 2020), ViM (Wang et al., 2022), ReAct (Sun et al., 2021), DICE (Sun & Li, 2022), SHE (Zhang et al., 2023), and FeatureNorm (Yu et al., 2023b), we adopt the same training scheme as in (Ming et al., 2023) for CIFAR-10 and CIFAR-100. Specifically, the models are trained using stochastic gradient descent with momentum 0.9 and weight decay $10^{-4}$. The initial learning rate is 0.1 and decays by a factor of 10 at epochs 100, 150, and 180. We train the models for 200 epochs on both CIFAR-10 and CIFAR-100. For the ImageNet benchmark, we adopt PyTorch's pre-trained ResNet-50 model with ImageNet-1k weights.

**Training contrastive models.**   For methods using SupCon loss (Khosla et al., 2020), such as KNN+ (Khosla et al., 2020) and SSD+ (Sehwag et al., 2021), we adopt the same training scheme as in (Ming et al., 2023). For CIFAR-10 and CIFAR-100 models, we use stochastic gradient descent with a momentum of 0.9 and a weight decay of $10^{-4}$. The initial learning rate to 0.5 and follows a cosine annealing schedule. We train the models for 500 epochs. The training-time temperature $\tau$ is set to be 0.1. For the ImageNet model, we use the checkpoint provided in (Sun et al., 2022b). For methods using CIDER, the models are trained using stochastic gradient descent with momentum 0.9 and weight decay $10^{-4}$. The initial learning rate is 0.5 and follows a cosine annealing schedule. We use a batch size of 512 and train the model for 500 epochs. The training-time temperature $\tau$ is set to be 0.1. We adopt the exponential-moving-average (EMA) for the prototype update (Li et al., 2020), with a momentum of 0.99 for CIFAR-10 and 0.5 for CIFAR-100. Training details of ImageNet-1k are included in Section 4.3.

**Evaluation configurations.**   We outline the configurations for methods that require appropriate hyperparameter selection, which has already been extensively studied in the literature.

- MSP uses the maximum softmax probability obtained from the logits. The method does not require any specific configuration.

- ODIN uses temperature scaling to calibrate the softmax score. We set the temperature $T$ to 1,000.

- Mahalanobis utilizes class conditional Gaussian distributions based on the low- and upper-level features obtained from a model. These distributions are then used to calculate the Mahalanobis distance.

- Energy derives the energy score, which includes the temperature parameter. We set the temperature to the default value of $T = 1$.

- ViM generates an additional logit from the residual of the feature against the principal space. We set the dimension of principal space to $D = 256$ for ResNet-18 and ResNet-34 and $D = 1024$ for ResNet-50.

- ReAct improves the energy score by rectifying activations at an upper limit, which is set based on the $p$-th percentile of the activations estimated on the in-distribution (ID) data. We set $p$ to the default value of $p = 90$.

- DICE utilizes logit sparsification to enhance the vanilla energy score, which is set based on the $p$-th percentile of the unit contributions estimated on the ID data. We set the sparsity parameter $p = 0.7$.

- SHE stores the mean direction of the penultimate layer features from correctly classified training samples. During inference, the Hopfield energy score is calculated as the dot product between the sample embedding and the class mean of the predicted class.

- FeatureNorm utilizes the norm of the feature map obtained from the chosen block in the neural network. We utilize the ImageNet-1k results as represented in Yu et al. (2023b),

and for CIFAR-10 and CIFAR-100, we run the experiments using block 4.1 as the selected block.

- `SSD+` applies the Mahalanobis score on the penultimate layer features obtained from a model pre-trained with the `SupCon` loss.

- `KNN+` and `CIDER` utilize the KNN score, which requires selecting a number of nearest neighbors $k$. Following the settings in Ming et al. (2023), we set $k$ to 100, 300, and 1,000 for CIFAR-10, CIFAR-100, and the ImageNet benchmark, respectively.

## B  VALIDATION METHOD FOR SELECTING TEST-TIME TEMPERATURE

To select the test-time temperature for our proposed hyperspherical energy score, we follow the validation method outlined in Hendrycks et al. (2019). We generate a validation distribution by corrupting in-distribution data with speckle noise, creating speckle-noised anomalies that simulate out-of-distribution data. After that, we compute the performance of hyperspherical energy at different test-time temperatures on the validation set and select the one that achieves the highest AUROC.

## C  ID CLASSIFICATION ACCURACY

Table 5 presents the in-distribution classification accuracy for each training dataset. We evaluate the classification accuracy of methods that involve learning hyperspherical representations, such as `KNN+`, `SSD+`, `CIDER`, and `Hyperspherical Energy`, by performing linear probes on normalized features, following the approach in Khosla et al. (2020). Our method shows competitive ID classification accuracy compared to the other existing methods, indicating it does not compromise the model's capability to distinguish samples between in-distribution classes.

Table 5: ID classification accuracy of each method on CIFAR-10, CIFAR-100, and ImageNet (%).

| Method | ID classification accuracy ↑ | | |
|---|---|---|---|
| | CIFAR-10 | CIFAR-100 | ImageNet |
| **Methods using cross-entropy loss** | | | |
| MSP | 94.21 | 75.03 | 76.15 |
| ODIN | 94.21 | 75.03 | 76.15 |
| Mahalanobis | 94.21 | 75.03 | 76.15 |
| Energy | 94.21 | 75.03 | 76.15 |
| ViM | 94.21 | 75.03 | 76.15 |
| ReAct | 93.95 | 74.43 | 74.89 |
| DICE | 93.92 | 74.38 | 73.72 |
| SHE | 94.21 | 75.03 | 76.15 |
| **Methods using hyperspherical representations** | | | |
| SSD+ | 94.75 | 75.42 | 79.10 |
| KNN+ | 94.75 | 75.42 | 79.10 |
| CIDER | 94.62 | 74.28 | 76.55 |
| Hyperspherical energy | 94.62 | 74.28 | 76.55 |

## D  RESULTS ON CIFAR-10 BENCHMARK

In this section, we present additional results and analysis on the CIFAR-10 benchmark, using the experimental settings described in Section A. As shown in Table 6, `Hyperspherical Energy` displays competitive performance compared to existing state-of-the-art methods. In particular, our method achieves FPR95 of 14.16%, which is similar to `CIDER`, which achieves 13.85%.

## E  VISUALIZATION ANALYSIS FOR LARGE-SCALE DATASET

Figure 4 presents the UMAP visualization of the learned embeddings derived from a subset of ImageNet-1k classes and larger-scale out-of-distribution (OOD) datasets. The class prototypes are

Table 6: OOD detection performance for CIFAR-10 (ID) with ResNet-18. Hyperspherical energy achieves competitive performance with state-of-the-art methods.

| Method | SVHN | | Places365 | | OOD Dataset LSUN | | iSUN | | Texture | | Average | |
|---|---|---|---|---|---|---|---|---|---|---|---|---|
| | FPR↓ | AUROC↑ | FPR↓ | AUROC↑ | FPR↓ | AUROC↑ | FPR↓ | AUROC↑ | FPR↓ | AUROC↑ | FPR↓ | AUROC↑ |
| **Methods using cross-entropy loss** | | | | | | | | | | | | |
| MSP | 59.81 | 91.25 | 62.57 | 88.69 | 45.43 | 93.80 | 55.20 | 92.03 | 66.61 | 88.50 | 57.92 | 90.85 |
| ODIN | 53.81 | 91.29 | 44.31 | 91.04 | 10.96 | 97.93 | 28.36 | 95.49 | 55.32 | 89.40 | 38.55 | 93.03 |
| Mahalanobis | 9.24 | 97.80 | 83.94 | 70.04 | 67.73 | 73.61 | 5.57 | 98.71 | 23.09 | 92.92 | 37.91 | 86.61 |
| Energy | 54.43 | 91.22 | 43.85 | 91.08 | 10.21 | 98.05 | 27.53 | 95.57 | 54.98 | 89.38 | 38.20 | 93.06 |
| ViM | 26.34 | 95.23 | 44.84 | 91.24 | 15.65 | 97.33 | 30.57 | 95.10 | 25.69 | 95.01 | 28.62 | 94.78 |
| ReAct | 48.21 | 92.20 | 48.11 | 90.97 | 23.03 | 95.96 | 22.02 | 96.38 | 48.90 | 91.19 | 38.05 | 93.34 |
| DICE | 65.34 | 89.66 | 50.44 | 89.81 | 3.95 | 99.21 | 34.98 | 94.87 | 59.22 | 88.50 | 42.79 | 92.41 |
| SHE | 64.29 | 88.31 | 70.13 | 80.70 | 8.00 | 98.56 | 55.27 | 90.78 | 58.10 | 87.99 | 51.16 | 89.27 |
| FeatureNorm | 8.79 | 98.27 | 76.75 | 79.84 | 0.16 | 99.92 | 37.67 | 94.17 | 29.96 | 94.08 | 30.67 | 93.26 |
| **Methods using hyperspherical representations** | | | | | | | | | | | | |
| SSD+ | 1.52 | 99.68 | 28.56 | 94.74 | 6.13 | 98.48 | 33.69 | 95.15 | 13.05 | 97.70 | 16.59 | 97.15 |
| KNN+ | 2.52 | 99.51 | 22.96 | 95.40 | 1.72 | 99.52 | 19.96 | 96.73 | 8.05 | 98.57 | 11.04 | 97.95 |
| CIDER | 3.46 | 99.37 | 31.57 | 94.50 | 2.59 | 99.37 | 15.97 | 97.28 | 15.66 | 97.58 | 13.85 | 97.62 |
| Hyperspherical energy | 3.89 | 99.28 | 32.59 | 94.14 | 3.05 | 99.29 | 16.02 | 97.20 | 15.27 | 97.64 | 14.16 | 97.51 |

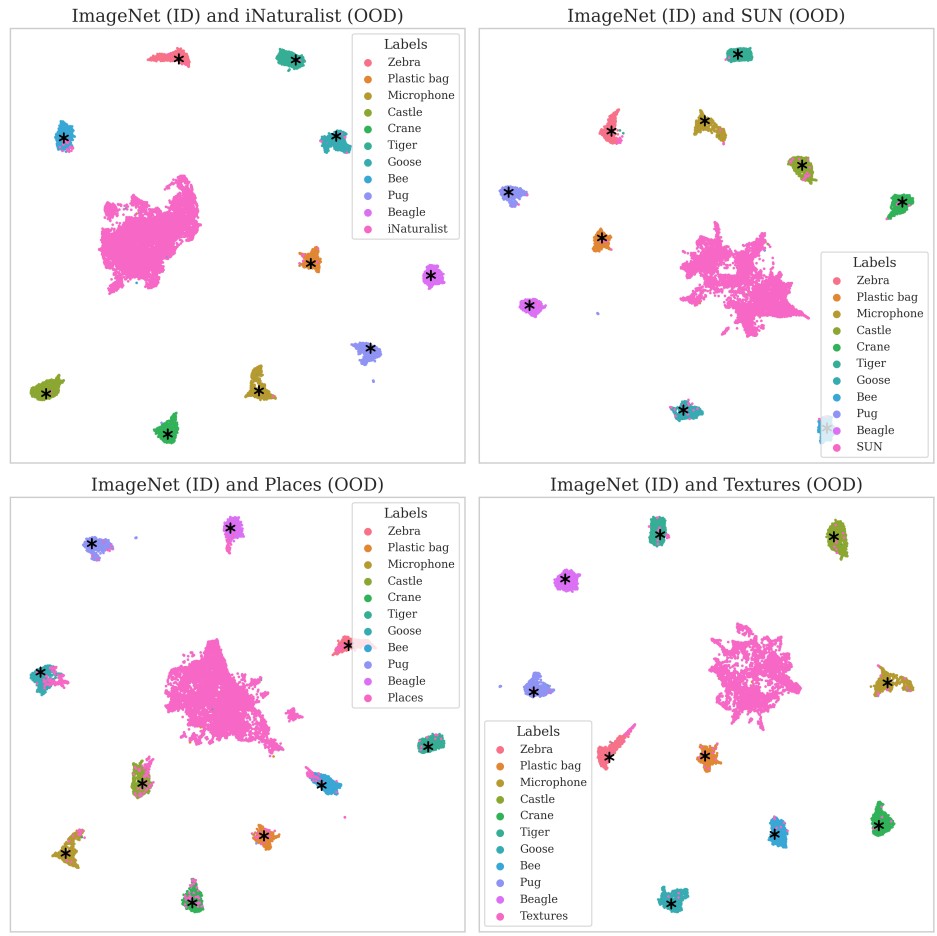

Figure 4: UMAP visualization of a subset of ImageNet classes and OOD datasets.

designated by a star symbol $\star$, while the OOD embeddings are distinguished by pink color. These visualizations indeed reveal compact representations, where each sample appears to be effectively drawn in towards its corresponding class prototype. A notable separation between in-distribution (ID) and OOD classes is also evident, suggesting that the OOD samples exhibit a high hyperspherical energy score.

# F   ID AND OOO DENSITY PLOTS

Figure 5 showcases the density plots of the hyperspherical score for In-Distribution (ID) data, using CIFAR-100, and Out-Of-Distribution (OOD) data, comprising SVHN, Places365, LSUN, iSUN, and Textures.

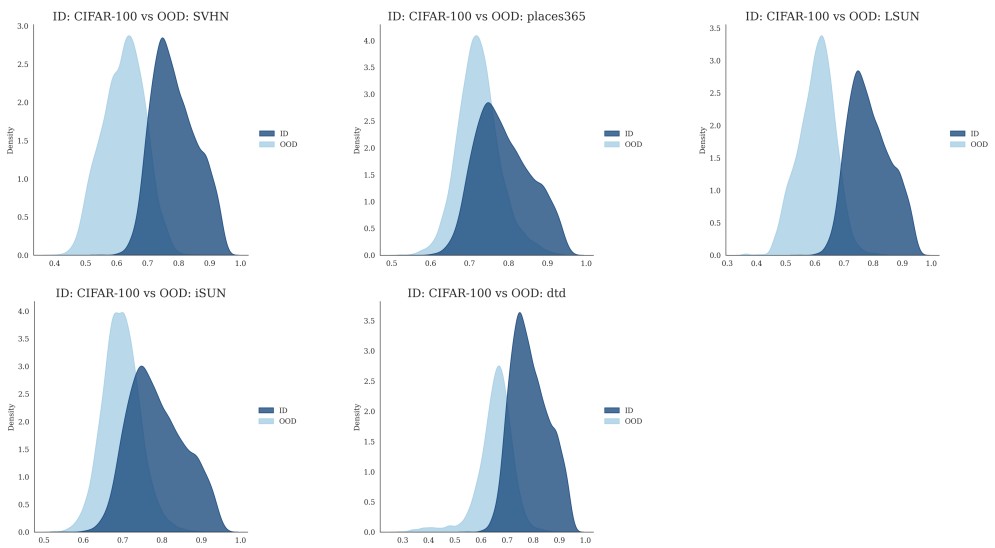

Figure 5: Density plots illustrating the distribution of hyperspherical scores.

