# OpenReview forum: "Out-of-Distribution Detection with Hyperspherical Energy"
_ICLR.cc/2024/Conference — Submitted to ICLR 2024_

### Official Review · Reviewer_A9ru · 2023-10-14

**Soundness:** 1 poor
**Presentation:** 2 fair
**Contribution:** 1 poor
**Rating:** 3
**Confidence:** 5

**Summary:**

This paper formulates hyperspherical embedding to enhance out-of-distribution (OOD) performance. From the training perspective, the authors train a hyperspherical embedding based on the von Mises-Fisher distribution based on the MLE objective. Then, they use the hyperspherical free energy as a test objective. The authors show the efficacy of the proposed method in out-of-distribution detection benchmarks with varying sizes. Furthermore, as the evaluation metric does not rely on the nearest neighbors, the method significantly excels in computational speed compared to alternative out-of-distribution methods that rely on the test scores.

**Strengths:**

(1) The method itself is simple and easy to understand.\
(2) The method is computationally appealing since it does not use nearest-neighbor search.

**Weaknesses:**

Although I am fond of the simple and efficient out-of-distribution detection algorithm as a baseline for further research, I have various concerns about this paper, including novelty, clarity, and significance.

Originality\
(1) Authors train the hyperspherical embedding via MLE loss, as done in previous papers [1,2]. \
(2) Authors focus on novelty in testing measures but this is just a proxy term for the negative log-likelihood. Although the perception of novelty is subjective, I also do not find this high-level concept of applying likelihood for OOD detection surprising or novel.

Clarity\
(1) While this paper's concept is simple, the paper lacks details.\
(2) For example, I found it confusing to directly compare CIDER, SIREN, and the proposed loss function since CIDER and SIREN impose additional regularization terms. \
(3) Or a basic specification for random variable z.

Significance\
(1) The paper frameworks the work as cross-entropy vs hyperspherical embeddings. But I do not think the comparison is fair. Classifiers trained by cross-entropy loss are widespread and easily collected in the wild. A strategy tailored to output OOD detection statistics from pre-trained classifiers via CE loss gets its merit due to its ubiquity. However, I do not feel the same applies to the hyperspherical embeddings. Instead, this is a pre-training strategy like [3,4]. Hence, I feel like the method should be **compared against various state-of-the-art** OOD detection methods when (image, label) information is given.\
(2) I find the gain of this method as a postprocessing strategy is limited. For example, [5] scores 94.28\%, [6] scores 95.06\%, and greatly outperforms the performance of this method despite being trained on the CE loss. While this method can benefit its performance by combining such methods, it is not sure in the current experiment results and should be verified via extra experiments.\
(3) Furthermore, I am also skeptical of hyperspherical embedding as a training strategy that learns better representation **in general**. A recently proposed benchmark of [7] compares CIDER [2] (which originates from the same training framework as this paper) to other strategies and shows its deficit in Near-OODs (e.g. SSB-Hard, NINCO) even compared to a simple rotation prediction strategy [8]. Moreover, it scores the worst among all compared to pretraining strategies. I think the bold assumption of this paper of showcasing hyperspherical embedding as an effective training loss should be verified by comparing other strategies (e.g. AugMix, RandAugment, RegMixup) on such challenging OOD, instead of comparing against only easy ones.\
(4) While I mainly focused on criticizing on ImageNet-1k dataset, the same reasoning can be applied to CIFAR-100 given the low scores in Table 1. of [7].

In summary,  given that this work is outperformed by various works that perform OOD detection on a classifier trained by CE or other strategies, I do not find any reason why should we train the classifier on hyperspherical embeddings (which is a very time-consuming task, especially for larger real-world datasets) and then perform OOD detection. Furthermore, I feel like this paper proposes a straightforward combination of existing works: hyperspherical embedding for training and energy function for OOD detection. Hence, I lean towards rejection.

**Questions:**

Questions\
(1) Are the training loss function of CIDER and the paper the same in the experiment? ID Acc of two is exactly the same in Table 5 but the loss function in the CIDER paper is different.\
(2) Does the proposed loss show efficacy in detecting near-OOD datasets? (e.g. SSB-Hard, NINCO in [7])\
(3) How does the proposed method perform against the common corruption?


Suggestions\
(1) To show the efficacy of the proposed method as the training strategy to learn **better representations**, consider comparison against training methods mentioned in the OpenOOD 1.5 [7] benchmark in more various OOD datasets not resorting to easier Far-OOD ones.\
(2) To show the efficacy of the proposed hyperspherical energy as a post-hoc inference strategy, consider comparison against Post-hoc inference methods mentioned in the weakness section and the OpenOOD 1.5 [7] benchmark.\
(3) Please specify z before use in the introduction section.


References\
[1] SIREN: Shaping Representations for Detecting Out-of-Distribution Objects, Neural Information Processing Systems 2020.\
[2] How to Exploit Hyperspherical Embeddings for Out-of-Distribution Detection? International Conference on Learning Representations 2023.\
[3] AugMix: a simple data processing method to improve robustness and uncertainty, International Conference on Learning Representations 2020.\
[4] RandAugment: Practical Automated Data Augmentation with a Reduced Search Space, Neural Information Processing Systems 2020.\
[5] Boosting Out-of-Distribution Detection with Typical Features, Neural Information Processing Systems 2022.\
[6] Extremely Simple Activation Shaping for Out-of-Distribution Detection, International Conference on Learning Representations 2023.\
[7] OpenOOD v1.5: Enhanced Benchmark for Out-of-Distribution Detection, arXiv 2023.\
[8] Using Self-Supervised Learning Can Improve Model Robustness and Uncertainty, Neural Information Processing Systems 2019.

---

> ### Author Response · Authors · 2023-11-21
> **Response to Reviewer A9ru**
>
> > Near-OOD datasets
>
> Thank you for the valuable suggestions regarding the evaluation of Near-OOD benchmarks. The table below presents a performance comparison of various methods on the suggested dataset NINCO (w.r.t. ImageNet-1k). This comparison highlights the efficacy of the Hyperspherical Energy method, which shows an overall improvement in performance compared to the Energy and CIDER methods.
>
> |                        | NINCO   |           |
> |:--------------------- |:------- | --------- |
> | **Method**            |  **FPR** | **AUROC** |
> | MSP                   | 76.37   | 79.94     |
> | Energy                |  77.63   | 79.69     |
> | CIDER                 |  72.63   | 78.17     |
> | Hyperspherical Energy |  **64.66**   | **84.68**     |
>
> > How does the proposed method perform against the common corruption?
>
> Thank you for your suggestion on testing against corrupted OOD data. In response to this, we've conducted extensive evaluations on CIFAR-10 and CIFAR-100 under AugMix and RandAugment augmentation scenarios on OOD datasets, as detailed in the tables below. These results clearly demonstrate the effectiveness of the Hyperspherical Energy method in dealing with harder OOD datasets (from common corruptions) by consistently achieving top performance compared to traditional Energy score and CIDER in various scenarios and datasets.
>
> **Under AugMix Augmentation**
>
> |                       | SVHN    |           | Places365 |           | LSUN    |           | iSUN    |           | Texture |           | Average |           |
> |:--------------------- | ------- |:--------- |:--------- | --------- | ------- | --------- | ------- | --------- |:------- |:--------- |:------- | --------- |
> | **Method**            | **FPR** | **AUROC** | **FPR**   | **AUROC** | **FPR** | **AUROC** | **FPR** | **AUROC** | **FPR** | **AUROC** | **FPR** | **AUROC** |
> | **CIFAR-100** |
> | Energy                | 72.33   | 86.38     | 81.85     | 76.97     | 70.17   | 84.59     | 76.01   | 81.16     | 87.93   | 74.18     | 77.66   | 80.66     |
> | CIDER                 | 25.03   | 94.90     | 77.88     | 74.62     | 16.65   | 96.29     | 68.27   | 81.59     | 43.90   | 90.41     | 46.35   | 87.56     |
> | Hyperspherical energy | 19.69   | 96.08     | 74.81     | 77.34     | 8.35    | 98.28     | 55.09   | 86.71     | 31.51   | 93.48     | 37.89   | 90.38     |
> | **CIFAR-10** |
> | Energy                | 68.44   | 86.20     | 78.97     | 79.60     | 65.64   | 86.13     | 70.53   | 83.12     | 83.92   | 76.91     | 73.50   | 82.39     |
> | CIDER                 | 24.11   | 95.08     | 71.73     | 78.55     | 22.31   | 94.83     | 64.50   | 82.30     | 42.04   | 91.14     | 44.94   | 88.38     |
> | Hyperspherical energy | 18.26   | 96.46     | 68.40     | 81.20     | 14.24   | 97.08     | 51.05   | 87.80     | 29.77   | 94.06     | 36.34   | 91.32     |
>
> **Under RandAugment Augmentation**
>
> |                       | SVHN    |           | Places365 |           | LSUN    |           | iSUN    |           | Texture |           | Average |           |
> |:--------------------- | ------- |:--------- |:--------- | --------- | ------- | --------- | ------- | --------- |:------- |:--------- |:------- | --------- |
> | **Method**            | **FPR** | **AUROC** | **FPR**   | **AUROC** | **FPR** | **AUROC** | **FPR** | **AUROC** | **FPR** | **AUROC** | **FPR** | **AUROC** |
> | **CIFAR-100** |
> | Energy                | 53.89   | 91.05     | 43.25     | 91.38     | 13.08   | 97.60     | 29.19   | 95.27     | 56.74   | 89.31     | 39.23   | 92.92     |
> | CIDER                 | 4.49    | 99.20     | 28.64     | 95.07     | 2.73    | 99.33     | 15.81   | 97.30     | 14.93   | 97.66     | 13.32   | 97.71     |
> | Hyperspherical energy | 5.16    | 99.07     | 29.82     | 94.70     | 3.31    | 99.26     | 16.24   | 97.19     | 14.80   | 97.68     | 13.87   | 97.58     |
> | **CIFAR-10** |
> | Energy                | 44.48   | 92.65     | 38.75     | 92.60     | 12.50   | 97.73     | 24.07   | 96.14     | 48.72   | 91.16     | 33.70   | 94.06     |
> | CIDER                 | 4.40    | 99.23     | 25.88     | 95.51     | 3.97    | 99.16     | 11.89   | 97.78     | 12.91   | 97.92     | 11.81   | 97.92     |
> | Hyperspherical energy | 5.16    | 99.10     | 27.43     | 95.13     | 4.67    | 99.06     | 12.94   | 97.69     | 13.58   | 97.87     | 12.76   | 97.77     |
>
> > Please specify $\mathbf{z}$ before use in the introduction section.
>
> Thank you for pointing that out! We have carefully reviewed the paper again to ensure that we introduce and describe the notations before using them.

---

> > ### Comment · Reviewer_A9ru · 2023-11-21
> >
> > Thank you for the response.\
> > While the authors resolve some of my concerns, I feel like the main concerns are not resolved since the response contains a partial response to my concerns. Hence, I stick to the current scores. Below are my further concerns on the author's rebuttal.
> >
> > 1. About near-OOD datasets: While I acknowledge the result on NINCO is beyond the performance of baseline methods in the OpenOOD benchmark, I would like to see the general picture including other ones.
> > 2. I do not find any results in common corruption. If the authors used the corruption operation (e.g. blur noise, digital, snow...), they should specify what severity they used. It just seems like the author just applied data augmentation for OOD detection.

---

### Official Review · Reviewer_fiAA · 2023-10-27

**Soundness:** 3 good
**Presentation:** 4 excellent
**Contribution:** 2 fair
**Rating:** 6
**Confidence:** 4

**Summary:**

This paper tackles the problem of ouf-of-distribution (OOD) detection by learning a representation that follows a
mixture of Von Mises-Fisher distributions. Then, on test time, they compute the Helmholtz free energy or the negative log partition function of the reconstructed logits with the class conditional prototypes and the embedding of the test sample. They run experiments on common image classification OOD detection benchmarks on CIFAR
and ImageNet datasets.

**Strengths:**

They propose a theoretically grounded parametric similarity score based on the representations learned under an energy-based model.

They achieve good experimental results on CIFAR and ImageNet for a ResNet model.

The paper is well-presented and easy to follow.

**Weaknesses:**

Their method requires special training that might be hard to transfer to any domain or any scale
(e.g., how to pick the best temperature parameter, etc). On the ImageNet dataset,
they are obliged to fine-tune only the last residual block of a pre-trained ResNet to achieve good results. I wonder if there are instabilities on the training of the entire network or if the generalization error is elevated.

Sufficient empirical investigation is lacking to justify some claims of the paper. How does the model perform on vision transformers for instance? Are the learned representations still useful for OOD detection?

Their contribution seems incremental w.r.t previous work, especially CIDER and SIREN. Differences w.r.t SIREN
are not discussed in Section 3.3.

Discussion on the limitations of the method is missing.

**Questions:**

1. Why did the authors freeze the first blocks of the backbone? If I understood correctly,
the network was first trained with a supervised contrastive objective and then trained with the loss function of Eq. (13).
2. I suggest the authors better discuss the differences of their method w.r.t SIREN.
3. I suggest running experiments with vision transformers to strengthen your empirical contributions and the relevance
of the proposed method. Or adjusting the claims accordingly.
4. Could the authors provide experimental details to obtain the results in Table 2?
Additionally, error bars and the number of tries on the results would be appreciated.
5. Could the authors provide error bars for their results by training on multiple random seeds? I believe this would strengthen their
empirical contributions.

---

> ### Author Response · Authors · 2023-11-21
> **Response to Reviewer fiAA**
>
> We sincerely thank the reviewer for the detailed review and constructive criticism. We also appreciate the reviewer's recognition of our paper's clarity and strong results. We address the question below in detail.
>
> > Q1. Why fine-tune the last residual block only?
>
> That's a great question. We performed fine-tuning on the last residual blocks, mostly to lower computational cost due to limited GPU resources. One can easily use our method and fine-tune on the entire backbone, without stability issues. On small-scale datasets such as CIFAR-10 and CIFAR-100, our training is indeed performed from scratch, on the entire network backbone.
>
> > Q2. Difference w.r.t. SIREN
>
> We highlight the major differences below:
>
> - SIREN uses maximum class-conditional likelihood (see **Table 1**), which is mathematically different from the hyperspherical energy score. Unlike SIREN, hyperspherical energy is theoretically sound for OOD detection, due to its log-likelihood interpretation (see Section 3.1 for details). Our OOD scoring function enjoys rigorous theoretical interpretation from a log-likelihood perspective, while SIREN's does not. Our derivation and interpretation of hyperspherical energy provided in Section 3.1 is an entirely new contribution, relative to SIREN.
> - SIREN primarily focuses on representation shaping loss during training time. In contrast, we focus on a novel test-time OOD detection scoring function. Our method is the first to establish the connection between the hyperspherical representations and Helmholtz free energy for OOD detection. From a training-time perspective, we also derive new insight into how the learning objective induces lower hyperspherical energy (Section 3.2), which directly connects to our proposed OOD scoring function.
>
> > Q3. Performance on vision transformers
>
> Thanks for the suggestion - these new experiments are underway and will be added to our updated manuscript.
>
>
> > Q4. Could the authors provide experimental details to obtain the results in Table 2? Additionally, error bars and the number of tries on the results would be appreciated.
>
> To provide clarity on the results in Table 2, we focused on measuring the time required to calculate the score for each sample. We obtained these results by averaging the time taken to calculate scores for samples in the ID dataset across each column. **The updated Table 2 now includes error bars**, indicating the variability in results across samples in the dataset.
>
>
> > Q5. Could the authors provide error bars for their results by training on multiple random seeds? I believe this would strengthen their empirical contributions.
>
> Excellent point. Indeed, we have provided error bars in **Figure 3**, where the results are based on multiple training over 3 random seeds.
>
>
> > Q6. Limitations
>
> A key strength of our method is its grounded interpretation from a likelihood perspective, as outlined in Lemma 3. This approach, however, necessitates training the model under the von Mises-Fisher (vMF) distribution. While our method achieves accuracy comparable to state-of-the-art models in certain scenarios, it's important to note that it is less common than models trained using traditional cross-entropy loss. This may limit its immediate applicability in environments where standard cross-entropy loss models are the norm. However, we believe that the trustworthiness of the models is worth an effort to train them in a certain way.

---

> > ### Comment · Reviewer_fiAA · 2023-11-21
> >
> > I appreciate the author's effort invested in the rebuttal. All my concerns were satisfactorily addressed, so I raise my score accordingly. I will keep an eye for the results on ViT and the discussion with the other reviewers.

---

> > > ### Author Response · Authors · 2023-11-21
> > > **thank you**
> > >
> > > Dear reviewer fiAA,
> > >
> > > Thank you for reading our rebuttal and increasing the score! We are glad to hear that the rebuttal resolved your concerns. Your feedback has certainly helped strengthen our manuscript :)
> > >
> > > Best,
> > > Authors

---

### Official Review · Reviewer_aQu1 · 2023-10-31

**Soundness:** 3 good
**Presentation:** 3 good
**Contribution:** 2 fair
**Rating:** 5
**Confidence:** 3

**Summary:**

This paper addresses the critical challenge of Out-of-Distribution (OOD) detection in machine learning models. It introduces a novel framework known as "Hyperspherical Energy" for OOD detection, which connects energy with hyperspherical representations, offering a rigorous perspective from a log-likelihood standpoint. The method is designed to provide a reliable and theoretically justified OOD detection indicator, addressing the limitations of energy-based scores derived from models trained with cross-entropy loss. Hyperspherical energy operates on latent embeddings, modeled as hyperspherical embeddings with constant norm, and is based on the negative log partition function of the Gibbs-Boltzmann distribution. The paper includes theoretical derivations and optimization techniques. Notably, the proposed method demonstrates competitive performance on OOD detection benchmarks, outperforming some baselines. The paper highlights the key distinctions and advantages of Hyperspherical Energy compared to conventional energy-based methods and non-parametric density estimation techniques. It is shown to achieve both strong empirical performance and theoretical soundness.

**Strengths:**

The proposed idea in this work is both novel and intriguing. The paper is commendably well-written, with a clear problem statement and easy-to-follow presentation. The authors not only provide solid theoretical proofs but also back their claims with experimental studies that demonstrate the promising performance of the proposed method.

**Weaknesses:**

The literature review in this paper exhibits a notable limitation as it primarily concentrates on a subset of OOD detection methods, while omitting recent approaches like [1,2,3,4]. Additionally, the experimental evaluation appears to lack comprehensiveness as it does not encompass recent OOD detection methods. Furthermore, a critical point of consideration is that the proposed method involves a training phase, whereas several OOD detection techniques necessitate minimal or no training, rendering them more practical and applicable in real-world scenarios.

[1] Huang, Rui, Andrew Geng, and Yixuan Li. "On the importance of gradients for detecting distributional shifts in the wild." Advances in Neural Information Processing Systems 34 (2021): 677-689.
[2] Behpour, Sima, et al. "GradOrth: A Simple yet Efficient Out-of-Distribution Detection with Orthogonal Projection of Gradients." arXiv preprint arXiv:2308.00310 (2023).
[3] Conor Igoe, Youngseog Chung, Ian Char, and Jeff Schneider. How useful are gradients for ood detection really? arXiv preprint arXiv:2205.10439, 2022.
[4] Andrija Djurisic, Nebojsa Bozanic, Arjun Ashok, and Rosanne Liu. Extremely simple activation shaping for out-of-distribution detection. arXiv preprint arXiv:2209.09858, 2022.

**Questions:**

The paper would benefit from a more elaborate discussion that highlights the distinct advantages of the proposed method compared to recent OOD detection studies, such as references [1,2,3,4]. Specifically, it would be valuable to clarify what sets this approach apart, especially considering the promising performance of ASH[4], which is both fast and does not necessitate any training.

In light of the growing interest in training-free OOD detection methods, it would be intriguing to explore the possibility of customizing your approach to require minimal or no training, enhancing its real-world applicability.

Including density plots that demonstrate the OOD and in-distribution densities using your method and the baseline approaches would provide a more comprehensive visual understanding of the results.

Additionally, it would be beneficial to delve deeper into the reasons behind the observed variations in your method's performance compared to the Energy score (Figure 2), particularly when it significantly outperforms the Energy score on datasets like SVHN and LSUN but demonstrates marginal improvements on other datasets such as Places 365. This elucidation would enhance the paper's clarity and insights.


[1] Huang, Rui, Andrew Geng, and Yixuan Li. "On the importance of gradients for detecting distributional shifts in the wild." Advances in Neural Information Processing Systems 34 (2021): 677-689.
[2] Behpour, Sima, et al. "GradOrth: A Simple yet Efficient Out-of-Distribution Detection with Orthogonal Projection of Gradients." arXiv preprint arXiv:2308.00310 (2023).
[3] Conor Igoe, Youngseog Chung, Ian Char, and Jeff Schneider. How useful are gradients for ood detection really? arXiv preprint arXiv:2205.10439, 2022.
[4] Andrija Djurisic, Nebojsa Bozanic, Arjun Ashok, and Rosanne Liu. Extremely simple activation shaping for out-of-distribution detection. arXiv preprint arXiv:2209.09858, 2022.- Could you please elaborate more and

---

> ### Author Response · Authors · 2023-11-21
> **Response to Reviewer aQu1**
>
> We thank you for the thorough and encouraging comments! We appreciate that you recognize the novelty and theoretical soundness of our work. We address your comments below in detail:
>
> > Distinct advantages of the proposed method compared to recent OOD detection studies
>
> That's an excellent question! The most significant advantage of our method lies in its **principled interpretation from a likelihood perspective (see Lemma 3)**. While related works [1, 2, 3, 4] have shown empirical promise, they lack a rigorous guarantee and cannot be viewed as likielihood-based test --- which is the fundamental crux for out-of-distribution detection.
>
> > Comparison with additional baselines
>
>
> We have additionally compared with these approaches, and provide the results below for your reference (on CIFAR-100 with ResNet-34 model). Our method outperforms the current best post-hoc method ASH by a significant margin, while enjoying the theoretical rigor.
>
> |                       | SVHN      |           | Places365 |           | LSUN     |           | iSUN      |           | Texture   |           | Average   |           |
> |:--------------------- | --------- |:--------- |:--------- | --------- | -------- | --------- | --------- | --------- |:--------- |:--------- |:--------- | --------- |
> | **Method**            | **FPR**   | **AUROC** | **FPR**   | **AUROC** | **FPR**  | **AUROC** | **FPR**   | **AUROC** | **FPR**   | **AUROC** | **FPR**   | **AUROC** |
> | GradNorm [1]          | 91.05     | 67.13     | 55.72     | **86.09** | 97.80    | 44.21     | 89.71     | 58.23     | 96.20     | 52.17     | 86.10     | 61.57     |
> | ExGrad [3]            | 77.29     | 81.45     | 78.70     | 78.62     | 74.93    | 81.69     | 75.61     | 80.83     | 81.44     | 78.78     | 77.59     | 80.27     |
> | ASH_B@65 - Energy [4] | 61.87     | 87.08     | 79.66     | 77.24     | 75.38    | 80.76     | 71.51     | 81.45     | 75.09     | 77.30     | 72.70     | 80.77     |
> | ASH_S@90 - Energy [4] | 38.00     | 93.24     | 81.77     | 72.92     | 75.69    | 78.41     | 67.10     | 82.72     | 64.08     | 83.23     | 65.33     | 82.10     |
> | Hyperspherical Energy | **17.81** | **96.39** | **76.74** | 76.01     | **8.46** | **98.25** | **57.92** | **85.98** | **31.83** | **93.24** | **38.55** | **89.97** |
>
> GradOrth [2] has not yet released their codebase. We will include it in our manuscript once it becomes available!
>
> [1] Huang, Rui, Andrew Geng, and Yixuan Li. "On the importance of gradients for detecting distributional shifts in the wild." Advances in Neural Information Processing Systems 34 (2021): 677-689.
>
> [2] Behpour, Sima, et al. "GradOrth: A Simple yet Efficient Out-of-Distribution Detection with Orthogonal Projection of Gradients." arXiv preprint arXiv:2308.00310 (2023).
>
> [3] Conor Igoe, Youngseog Chung, Ian Char, and Jeff Schneider. How useful are gradients for ood detection really? arXiv preprint arXiv:2205.10439, 2022.
>
> [4] Andrija Djurisic, Nebojsa Bozanic, Arjun Ashok, and Rosanne Liu. Extremely simple activation shaping for out-of-distribution detection. arXiv preprint arXiv:2209.09858, 2022.
>
> > ID-OOD density plot
>
> Great suggestion! We have included such a plot in our revised draft. Please see **Appendix F** for details.
>
> > Performance on different datasets
>
> The high FPR95 for places365 might be due to the artifact in the dataset itself, rather than the methodology. Despite its common use in literature, the test dataset can be flawed since many images can be distributionally close to ID data. For example, recent work [5] pointed out that Places have some overlapping classes with the ImageNet-1k, which may lead to high FPR95 in evaluation.
>
> To investigate this hypothesis for CIFAR-100, we analyze the FPR95 for each category in Places365. We found that more than half of the 365 types of images have FPR95 of at least 79%. This suggests that there’s a lot of overlap between ID and OOD data, which leads to high rates of false positives. On closer inspection, we saw that certain types of images, like natural landscapes and room scenes, are more likely to have this problem. For example, in the places365 dataset, images in the "television_room" category often get mixed up with "television", "couch", or "table" classes from CIFAR-100 because those objects are in the pictures. This confusion leads to FPR95 to be as high as 97%. Another example is the "storage_room" images in places365. They get classified as "table" or "wardrobe" if those items are in the picture, resulting in an FPR95 of 79%.
>
> [5] Bitterwolf et al., In or Out? Fixing ImageNet Out-of-Distribution Detection Evaluation, ICML 2023.

---

> > ### Comment · Reviewer_aQu1 · 2023-11-21
> >
> > Thanks for providing the rebuttal!
> > I have no question any more and would consider your rebuttal in my final decision/score.

---

> > > ### Author Response · Authors · 2023-11-21
> > > **thank you!**
> > >
> > > Dear reviewer aQu1,
> > >
> > > Thanks for taking the time to read our response and sending us the message here! We really appreciate your effort in the midst of this busy season.
> > >
> > > take care,
> > > Authors

---

### Official Review · Reviewer_uYkE · 2023-10-31

**Soundness:** 3 good
**Presentation:** 2 fair
**Contribution:** 2 fair
**Rating:** 5
**Confidence:** 3

**Summary:**

The authors study out-of-distribution (OOD) detection for classification problems.

They propose hyperspherical energy, an OOD detection method that combines energy-based OOD detection (Liu et al., 2020) with recent hyperspherical-based approaches such as CIDER (Ming et al., 2023). Compared to CIDER, the proposed method utilizes a parametric score function (instead of a kNN-based one).

The proposed method is evaluated with CIFAR-10, CIFAR-100 and ImageNet-1k as ID datasets, and SVHN, Places365, LSUN, iSUN and Textures as OOD datasets. It achieves better (CIFAR-100 and ImageNet, Table 3-4) or similar (CIFAR-10, Table 6) FPR95/AUROC compared to CIDER and other common strong baselines.

**Strengths:**

The proposed method makes sense overall. Combining hyperspherical-based and energy-based methods intuitively seems like a good idea.

The proposed method achieves good performance compared to both the hyperspherical-based and energy-based approaches.

**Weaknesses:**

The paper could be a bit more well-written overall (see "Minor things" in Questions below).

I found it somewhat difficult to follow Section 3.1 and 3.2, I think that the proposed method could be described more clearly. I think it would help if more background on CIDER (Ming et al., 2023) was provided before Section 3.

The experimental results are not analyzed in much detail. The OOD detection performance (FPR95/AUROC) is evaluated _relative_ to other methods, but nothing is said about the performance in _absolute_ terms. For example in Table 3, is an average FPR95 of 38.5 actually good? Why is there such a big performance difference between Places365 and LSUN? In which cases does the model fail to distinguish between ID and OOD examples? Why?

**Questions:**

1. What can be said of the _absolute_ OOD detection performance of the proposed method? For example in Table 3, is an average FPR95 of 38.5 good? Does the method actually perform well? In the Introduction you write _"The ability to detect OOD samples is crucial for ensuring the reliability and safety of machine learning models, especially in real-world applications where the consequences of misidentifying OOD samples can be severe"_, does the method perform well enough for this important task?

2. Can anything be said about common failure cases? For example, why is there such a big performance difference between Places365 and LSUN?

3. Does the proposed method have any limitations?

4. Could the proposed OOD detection approach be extended to regression problems (I am mostly just curious)?


Minor things:
- In equation (13), is $\mathcal{L}$ the loss from equation (12)? ($\mathcal{L}$ is not really defined anywhere?)
- I was confused by how $\cdot$ is used both to multiply scalars (e.g. in (1) and (5)) and vectors (e.g. $\mu_c \cdot z$ in (6) and (7)).
- Introduction, "Hyperspherical energy vs. energy" bullet point, "Liu et al. (2020) employ cross-entropy loss and derive energy from the unconstrained Euclidean space": cross-entropy loss --> a/the cross-entropy loss?
- Introduction, "Hyperspherical energy vs. energy" bullet point, "constrained hyperspherical distribution": distribution --> distributions?
- Introduction, "Hyperspherical energy vs. energy" bullet point, "our method enjoys ... theoretical property": This sounds a bit odd?
- Section 2, "EBM (Lecun et al., 2006) captures...": EBM --> An EBM?
- Section 2, "a connection between EBM and discriminative models": EBM --> EBMs?
- Section 2, "Limitation of Liu et al. (2020)" paragraph: This description is perhaps a bit too short/condensed?
- Section 3.3: "SupCon loss’s formulation" --> "SupCon's loss formulation"?
- Section 3.3: Add a reference for SupCon?

---

> ### Author Response · Authors · 2023-11-21
> **Response to Reviewer uYkE**
>
> We sincerely appreciate your positive feedback and insightful comments. Your acknowledgment of our work's clarity and unique contribution means a lot to us. We address the questions below in detail.
>
> > Q1. What can be said of the absolute OOD detection performance of the proposed method?
>
> Thanks for the insightful question. We believe it's meaingful to look at both the absolute and relative performance. In particular, Table 3 is based on a challenging setting of CIFAR-100 as ID. As you can see, early method such as MSP can have average FPR95 as high as 83.90%. This signifies the non-trivialness of the task itself. This can be attributed to two factors. First, the ID accuracy on CIFAR-100 is suboptimal (75.03\% as we report in Appendix Table 5), which means the model does not have a strong inherent characterization of the ID data within its representation. Secondly, the ID data and some OOD data can share spurious correlation (e.g., blue sky background), which further exacerbate the detection difficulty.
>
> Noticeably, in this challenging setting, our method can reduce the FPR95 from the earliest 83.9% to now 38.55% on this challenging task, which we believe is significant progress already. Admittedly, there is still large room for improvement in future research. From a practical point of view, we do desire as low absolute FPR95 as possible. That would be the dream of all researchers in this field, as make more progress :)
>
> > Q2. Common failure cases
>
> The high FPR95 for places365 might be due to the artifact in the dataset itself, rather than the methodology. Despite its common use in literature, the test dataset can be flawed since many images can be distributionally close to ID data. For example, recent work [1] pointed out that Places have some overlapping classes with the ImageNet-1k data, which may lead to high FPR95 in evaluation.
>
> To investigate this hypothesis for CIFAR-100, we analyze the FPR95 for each category in the places365 dataset. We found that more than half of the 365 types of images have FPR95 of at least 79%. This suggests that there’s a lot of overlap between ID and OOD data, which leads to high rates of false positives. On closer inspection, we saw that certain types of images, like natural landscapes and room scenes, are more likely to have this problem. For example, in the places365 dataset, images in the "television_room" category often get mixed up with "television", "couch", or "table" classes from CIFAR-100 because those objects are in the pictures. This confusion leads to FPR95 to be as high as 97%. Another example is the "storage_room" images in places365. They get classified as "table" or "wardrobe" if those items are in the picture, resulting in an FPR95 of 79%.
>
> > Q3. Limitations
>
> A key strength of our method is its grounded interpretation from a likelihood perspective, as outlined in Lemma 3. This approach, however, necessitates training the model under the von Mises-Fisher (vMF) distribution. While our method achieves accuracy comparable to state-of-the-art models in certain scenarios, it's important to note that it is less common than models trained using traditional cross-entropy loss. This may limit its immediate applicability in environments where standard cross-entropy loss models are the norm. However, we believe that the trustworthiness of the models is worth an effort to train them in a certain way.
>
> > Q4. Extension to regression
>
> That's a very interesting question! It may be possible to extend from classification to regression setting. For example, previous work [2] showed the feasibility for regression by optimizing outputs as an interpolation between two prototypes on the hypersphere. It's worth noting that the loss function employed in [2] was different from ours, so we need more conclusive investigation in the future to determine the feasibility.
>
> [1] Bitterwolf et al., In or Out? Fixing ImageNet Out-of-Distribution Detection Evaluation, ICML 2023.
> [2] Mettes et al., Hyperspherical Prototype Networks, NeurIPS 2019
>
> > **Minor suggestions on writing**
>
> We have fixed those writing glitches - thank you for the careful read!

---

> > ### Comment · Reviewer_uYkE · 2023-11-22
> > **Response to rebuttal**
> >
> > (Sorry for my late response. I have struggled to find enough time to both write responses as an author, and participate in the discussions as a reviewer)
> >
> > I have read the other reviews and all responses.
> >
> > The response to my question 1 and 2 is quite interesting. The new results in the response to Reviewer aQu1 are good additions.
> >
> > However, Reviewer A9ru raises some good points. In particular, Significance point (1) - (3) in Weaknesses.
> >
> > I am still borderline on this paper, I will keep my score of "5: marginally below" for now.

---

### Meta-Review · Area_Chair_FZKg · 2023-12-11

**Metareview:**

The paper introduces a novel approach to out-of-distribution (OOD) detection by integrating hyperspherical and energy-based methods into a parametric score function. It demonstrates strong performance in empirical benchmarks. Some reviewers praised the paper for the general presentation quality, found the methodological approach novel and theoretically well-substantiated, and noticed the strong performance of the method. However, other reviewers raised concerns about the novelty and found the contribution incremental, critiqued the clarity of the presentation of the methodology details, found limitation in the literature review and raised concerns about practical applications, as the proposed method involve a training phase.

**Justification For Why Not Higher Score:**

Most reviewers have identified significant limitations in the paper leading to a consensus that the paper falls below the acceptance threshold.

**Justification For Why Not Lower Score:**

N/A

---

### Decision · Program_Chairs · 2024-01-16

Reject